# Evaluating a Telephone and Home Blood Pressure Monitoring Intervention to Improve Blood Pressure Control and Self-Care Behaviors in Adults with Low-Socioeconomic Status

**DOI:** 10.3390/ijerph20075287

**Published:** 2023-03-28

**Authors:** Komlanvi S. Avegno, Kristina B. Roberson, Elijah O. Onsomu, Michelle F. Edwards, Eric L. Dean, Alain G. Bertoni

**Affiliations:** 1Division of Nursing, School of Health Sciences, Winston-Salem State University, 601 S. Martin Luther King, Jr Dr., Winston-Salem, NC 27110, USA; 2Triad Adult and Pediatric Medicine, 1002 S. Eugene Street, Greensboro, NC 27406, USA; 3Dean Internal Medicine, 1409 Yanceyville St., Ste C, Greensboro, NC 27405, USA; 4School of Medicine, Wake Forest University, 1 Medical Center Blvd, Winston-Salem, NC 27157, USA

**Keywords:** BP control, low socioeconomic status, self-care behaviors, telehealth intervention, home blood pressure monitoring

## Abstract

Hypertension (HTN) affects nearly 75 million in the United States, and percentages increase with low socioeconomic status (SES) due to poor access to, and quality of, care, and poor self-care behaviors. Federally Qualified Health Centers (FQHCs) employ evidence-based strategies, such as telehealth interventions, to improve blood pressure (BP) control in under-resourced communities, yet a southeastern FQHC could achieve a BP control rate of only 27.6%, well below the Health People 2020 goal of 61.2%. This pilot project used a pre/post, matched-cohort design to evaluate the effect of a telehealth intervention on BP control and self-care behaviors. Secondary outcomes included self-efficacy and perceived stress. Frequency and percentage, Wilcoxon signed-rank, and McNemar tests were used for statistical analysis of results from a convenience sample of 27 participants. Baseline HTN management guidance that incorporated home blood pressure monitoring (HBPM) was reinforced through telephone counseling every two weeks. Although BP control was not achieved, average scores for systolic and diastolic blood pressures decreased significantly: 13 mm Hg (*p* = 0.0136) and 5 mm Hg (*p* = 0.0095), respectively. Statistically significant differences were also seen in select self-care behaviors. Greater BP reduction aligned with higher self-efficacy scores and call engagement. Overall, telephone counseling and HBPM were feasible and effective in reducing BP and increasing self-care behaviors. The inability to control BP may be attributable to under-recognition of stress, lack of medication adherence/reconciliation, and underutilization of guideline-based prescribing recommendations. Findings elucidate the potential effectiveness of a sustainable telehealth intervention to improve BP in low-SES populations.

## 1. Introduction

Hypertension (HTN) is one of the most common chronic conditions in the United States, affecting approximately 75 million, or 33% of the adult population [1]. By 2030, an estimated 41% of US adults will have HTN, an almost five-fold increase over 2013 estimates [2]. Complications include macro and microvascular problems, such as stroke, coronary heart disease, cardiomyopathy, peripheral vascular disease, retinopathy, nephropathy, and neuropathy [2]. Treatment places a huge burden on the US economy, with direct and indirect costs for 2013 estimated at $329.7 billion [3].

The association between socioeconomic (SES) status and the risk for, and prevalence of, HTN is clear and strong [4]. SES is based on income, education, employment, and social position [5]. The higher rate of HTN in low-SES populations can be explained by a lack of health insurance, the inability to meet copay or out-of-pocket expenses for healthcare visits, poor dietary choices, poor health literacy, low self-efficacy, poor living conditions, unsafe neighborhoods, and/or no sidewalks and parks for walking and other physical activities. Individuals with a low SES face barriers to accessing care, receive poorer quality of care, and experience worse health outcomes. Income and educational attainment are two SES factors that demonstrate HTN disparity. According to America’s Health Rankings [6], the overall rate of HTN is higher in adults making <$24,999 and non-high school graduates. Overall, these disparities and inequities induce emotional stress, which can evoke a physiologic response mediated by the sympathetic nervous system. Stress spikes blood pressure, and when stress is chronic, it fails to return to resting levels, increasing circulating levels of catecholamines and cortisol levels [7].

Federally Qualified Health Centers (FQHCs) are uniquely positioned to improve the quality and outcomes of care through evidence-based practice among low-SES populations. FQHCs are federally funded nonprofit clinics, that provide primary care services to medically underserved areas and populations regardless of the ability to pay. They use federal poverty guidelines as eligibility criteria, thereby serving mostly the low-SES populations. Goldman et al. [8] found that FQHCs demonstrated performance equal to, or better than private practice primary care physicians on quality measures, despite serving patients with more chronic diseases and socioeconomic complexity. According to the Human Resources and Services Administration (HRSA) [9], a FQHC practice site cared for a population of which 97.72% had an income 200% lower than the federal poverty level. It is located in a city where 19.9% of the residents live in poverty [10], higher than the national rate of 12.7% [11]. Nearly 11.62% of the patients were uninsured, and 77.19% had Medicaid insurance [9]. HTN was the most common medical condition; 34.56% of the patients presented with it compared to 19.16% with diabetes and 10.15% with asthma [9]. Low self-care behaviors likely contributed to the low rates of controlled HTN (M. Edwards, personal communication, 20 July 2018). 

Self-care behaviors and self-efficacy have been shown to contribute to successful HTN management and a reduced risk of complications. Effective self-care behaviors reduce hospitalizations and improve the quality of life [12]. However, despite evidence that lifestyle changes and adhering to treatment recommendations can control BP, prevent hospitalization, and decrease mortality, many patients do not practice self-care behaviors, adhere to antihypertensive regimens, follow dietary advice, or tend to physical activity recommendations [13]. Helping them understand that HTN is a manageable condition increases their self-efficacy, or confidence that they can participate in a given behavior and improve BP control. Studies associate self-efficacy with self-reported and objective measures of adherence to medication regimens in addition to participation in physical activity and weight management. Self-efficacy has been found to predict self-care behaviors [14]. 

Healthy People 2020 set a target for adults with HTN: 61.2% would have their BP under control [15]. Internal monitoring at the practice site revealed that among adult patients seen in 2016 who had a diagnosis of HTN, only 27.6% met the criteria for BP control (C. Carter, personal communication, 10 January 2017), which is less than half of the goal [15]. 

The practice site for the project described here identified the need to find an effective intervention to improve BP control. According to the 2017 American College of Cardiology (ACC)/American Heart Association (AHA) Guideline for the Prevention, Detection, Evaluation, and Management of High BP in Adults (hereafter, 2017 ACC/AHA Guideline [16]), telephone interventions can be used to inform patients and family about medical instructions and encourage self-care goals. The guideline further recommends home blood pressure monitoring (HBPM) [16], an evidence-based recommendation that can be administered and monitored by the FQHC site. The telephone intervention recommending HBPM can be incorporated into routine practice to promote self-care behaviors in low-SES adults, leading to BP control.

Telehealth is defined as the remote delivery of healthcare services using information and communication technology. It includes a growing variety of applications and services, including two-way video, phone calls, email, the internet, smart phones, and wireless tools [16]. This emerging, viable intervention model aims to engage adult patients in managing their complex chronic conditions [17]. A telephone intervention allows healthcare providers to reinforce knowledge about HTN and can be used as a best practice to promote BP control and encourage self-care [18]. It has proven effective among the socially disadvantaged and is recommended by the US Preventive Services Task Force to help reduce cardiovascular risk factors, such as HTN [5]. 

The telehealth intervention of interest in this project is delivered by the provider via telephone call and incorporates HBPM. In addition to the 2017 ACC/AHA Guideline [16], a meta-analysis and systematic review [19], and a quasi-experimental study [20] found that telephone and HBPM interventions produced clinically or statistically significant improvements in BP control. In the first [19], the telephone intervention decreased mean systolic BP (SBP) by 3.74 mm Hg and mean diastolic BP (DBP) by 2.37 mm Hg, compared to usual care. While these findings were not statistically significant, the associated 8% reduction in stroke mortality and 5% reduction in mortality from coronary heart disease (CHD) have clear clinical relevance [21]. Moreover, a 2 mm Hg decrease in DBP reduced the risk of CHD by 6% and risk of stroke and transient ischemic attack (TIA) by 15% [22]. A 12 week study reviewed in [19] revealed both a statistically and clinically significant decrease in mean SBP, from 140 to 133 mm Hg (*p* = 0.001), and mean DBP from 84 to 80 mm Hg (*p* = 0.005) in the intervention group. In the control group, mean SBP decreased from 138 to 134 mm Hg and mean DBP from 83 mm Hg to 82 mm Hg. Secondary outcomes were self-efficacy, medication adherence, dietary changes, and physical activity, but data on these measures were not available for all the studies reviewed [19]. 

The second study [20] also found statistically significant BP reductions. Mean SBP values in the intervention group decreased from 153 to 130 mm Hg (*p* < 0.0001), and mean DBP values decreased from 89 to 76 mm Hg (*p* < 0.0001). In the usual-care group, mean SBP values decreased from 156 to 149 mm Hg (*p* < 0.05), and mean DBP values decreased from 90 to 86 mm Hg (*p* < 0.05). These reductions were also clinically significant: for every 10 mm Hg reduction in SBP, all-cause mortality related to major cardiovascular disease events drops 13% [23]. At the final office visit, the usual-care group had poor BP control: 81% had out-of-range SBP values, and 62% had out-of-range DBP values. In contrast, only 26% and 8% of the home-based telemedicine group had out-of-range SBP and DBP values, respectively. As a secondary outcome, 23% of the usual-care group had their drug regimen changed as compared to 70% of the home-based telemedicine group.

### 1.1. Purpose Statement

This pilot study implemented a telehealth intervention at a FQHC, and evaluated its effect on BP control and self-care behaviors in adults aged 18–55 years with a low SES. The primary outcomes were BP control and self-care behaviors; the secondary outcome was self-efficacy. The goal was to provide an intervention that could be easily and effectively adapted to clinical practice to improve healthcare delivery at the local site. Few studies have focused on using a telehealth intervention to achieve BP control in a FQHC setting. Additionally, the practice site had not targeted HTN patients for telehealth intervention. The primary question addressed was: How does a 12-week telehealth intervention affect BP control and self-care behaviors in adults aged 18–55 years with a low SES, compared to pre/post matched-cohort receiving usual care of routine 3-month office visits?

### 1.2. Theoretical Framework

The ACE STAR Model of Knowledge Transformation and the Information-Motivation-Behavioral Skills (IMB) model served as theoretical frameworks. The ACE STAR model [24] systematically organizes evidence-based processes and approaches, to facilitate their translation into practice. After a literature review, the evidence is synthesized with other knowledge, translated into practice, and evaluated for effectiveness. 

The IMB model [25], used repeatedly with chronic diseases [19], uses office and telephone counseling to provide participants with information, in this case, regarding HBPM, medication management, diet, physical activity, and stress management. Motivation is validated through measurements of self-efficacy on the understanding that it increases confidence and, therefore, self-efficacy. Behavioral skills are measured by self-care behavior based on the idea that self-efficacy is a predictor of and increases self-care behavior. Essentially, counseling, self-efficacy, self-care behavior, and HBPM interact to bring about behavioral change and BP control.

## 2. Methods

### 2.1. Design

This pilot and feasibility project used a pre/post, matched-cohort design. 

### 2.2. Setting and Sample

The practice site was a FQHC in a southeastern US county, designated as a priority area to address self-care behaviors and in a priority zip code to address poverty as a determinant of health. 

Convenience sampling was used with a priori project enrollment power analysis to determine that a minimum sample size of 35 was needed to achieve at least 0.80 power to detect a significant effect of a telehealth intervention on BP, with two-tailed significance set at 0.05. To account for 20% attrition, the required sample size was 42. A recruitment flyer was given to all patients at check-in and posted in heavy traffic areas, such as exam and rest rooms. Potential subjects were also identified by practice-site staff via the electronic health record system and self-referral via phone tree, the practice-site Facebook page, and word of mouth.

Inclusion criteria consisted of: (1) HTN diagnosis for at least six months according to the ICD-10 listing in the medical record; (2) uncontrolled BP of ≥140/90 mm Hg [26] based on the average of the three most recent measurements on ≥2 occasions; (3) self-reported access to a cellphone and/or landline; (4) aged 18–55 years; and (5) self-reported ability to read, understand, speak, and write English.

Exclusion criteria consisted of: (1) any current diagnosis or history of heart failure, myocardial infarction, end-stage kidney disease, or pregnancy per the ICD-10 listing in the medical record; and (2) any severe cognitive impairment. The rationale for excluding patients with the stated health conditions is based on the risk for complications, such as cardiogenic shock, arrhythmia, and cardiac arrest with heart failure and myocardial infarction; heart failure with end-stage kidney disease; and pre-eclampsia and HELLP syndrome with pregnancy, which must be managed by an obstetrician [27]. Subjects with severe cognitive impairment are excluded due to the self-care component of the intervention. 

### 2.3. Intervention

This intervention had two main components: a baseline office visit and scripted phone calls at two-week intervals. The face-to-face office visit encompassed informed consent, completion of baseline questionnaires, and phone-number verification. Counseling was provided using the Krames Staywell educational booklet, *Managing High Blood Pressure*. Content discussed included BP etiology, measurement, and monitoring, risk factors, medications, and the value of a healthy diet and lifestyle changes. Participants were also given and taught how to use an Omron Bp742n 5 series BP monitor, validated by the European Society of Hypertension International Protocol [28]. They received a BP log and were advised to record their BP once every day at the same time, not at night [16]. In the event that the device failed during the intervention, participants were instructed to measure their BP at a Walmart, CVS, and/or Walgreen’s store, using a BP kiosk. Patients could also come into the office, and a nurse could check their BP at no cost. 

Telephone calls were placed from the practice site every two weeks between 10:00 a.m. and 5:00 p.m. They were structured as scheduled calls, scheduled calls with referrals, and unscheduled calls. Scheduled calls followed the telephone-intervention script and focused on home BP measurement, medication adherence, physical activity, and dietary and stress management. Project personnel asked closed-ended questions requiring a yes or no response, and followed with education reinforcement, directing participants to an American Heart Association recommended booklet, as needed, entitled *Managing High Blood Pressure*. 

If participants reported elevated home BP readings or any other clinical problems during the scheduled calls, asked questions, initiated discussions, or had concerns outside of the script content, the call was categorized as a scheduled call with referral. An electronic medical record (EMR) task was sent to the appropriate clinical team member. Concerns regarding BP measurement, medication adherence, and physical activity were sent to the triage-registered nurse and/or primary care provider. Concerns regarding diet or stress were sent via EMR task to the registered dietician or behavioral health clinician, respectively. Participants were also given the option to be transferred to the respective voicemails to leave a message. 

Unscheduled calls were initiated by patients [20] and managed by the triage nurse. They captured clinical problems and other health concerns the patients wanted to discuss.

The 2017 ACC/AHA Guideline served as a reference for management [16]. If two SBP readings were >140 and/or DBP readings >90, an EMR task was sent to the triage-registered nurse for follow-up with the participant. If two SBP readings were >180 and/or DBP readings >120, participants were advised to come to the office immediately. A verbal notification was sent to the provider to expect a direct office visit. Many hypertensive patients with severe BP elevation have withdrawn from, or are noncompliant with, antihypertensive therapy and do not have clinical or laboratory evidence of acute target organ damage [16]. They should not be considered a hypertensive emergency, but should be treated by reinstitution or intensification of antihypertensive drug therapy and anxiety, as applicable. They need not be referred to the emergency department for immediate reduction in BP or hospitalized [16].

If participants had at least two SBP readings >180 and/or DBP readings >120, could not come to the office, and were experiencing symptoms, such as dizziness, chest pain, shortness of breath, back pain, numbness/weakness, change in vision, difficulty speaking, and severe headache [29], they were advised to dial 911 and schedule a follow-up visit after the ambulance technician or emergency department evaluation. The provider would receive immediate verbal notification of the participant’s status and disposition, and send an EMR task to ensure proper documentation. Scheduled intervention calls would resume. 

In the event that participants did not answer, a voice message was left. If they did not return the call within one hour, a second call was placed. If they remained unreachable, another voice message was left, and a third call was placed the next day between 10:00 a.m. and 12:00 p.m. If they remained unreachable, subsequent calls were placed at two-week intervals. 

During the end of project face-to-face office visit at 12 weeks, participants completed the post-intervention questionnaires, and trained personnel measured their BP. It lasted no more than 32 min. Trained personnel collected the rest of the data, which could be abstracted from the medical record during chart review. Participants received a $10 gift card and the HBPM device Omron Bp742n 5 series at the baseline visit and another $10 gift card during the post-intervention face-to-face office visit. Those who withdrew during the project received no second gift card, but were not otherwise penalized for withdrawal. 

### 2.4. Outcome Measures

This project measured two primary outcomes, BP control and self-care behaviors, and one secondary outcome, self-efficacy. Other captured items included clinical and sociodemographic data, patient and staff experience, and perceived stress.

BP measurement followed the 2017 ACC/AHA Guideline [16]. Control was assessed as attaining and maintaining <130/80. An SBP between 130–139 mm Hg or a DBP between 80–89 mm Hg constituted stage I HTN, and stage II was SBP > 140 mm Hg or DBP > 90 mm Hg. Hypertensive crisis was SBP > 180 mm Hg and/or DBP > 120 mm Hg. Trained clinic staff measured BP using calibrated dinamap equipment. If this automatic, noninvasive monitor malfunctioned, or the reading needed confirmation, manual readings were taken with a sphygmomanometer and stethoscope.

Self-care behaviors were defined as decisions and actions taken to regulate personal functioning in the interest of well-being [30]. The Hypertension Self-Care Activity-Level Effect (H-SCALE) was used to measure HTN self-care behaviors. This tool is consistently used with the hypertensive population, and the author gave us permission to use it. The H-SCALE is divided into six subscales that assess prescribed self-care activities: (1) medication adherence; (2) diet; (3) physical activity; (4) smoking; (5) weight management; and (6) alcohol intake [31]. It is designed specifically for use in primary care settings, and its 31 questions take 10–15 min to complete. The subscales have acceptable Cronbach’s alpha reliability: (1) medication adherence (α = 0.93); (2) diet (α = 0.71); (3) physical activity (α = 0.82); (4) weight management (α = 0.90); and (5) alcohol use (α = 0.88) [31]. All items were rated on a 7-point scale, except weight management, which was rated on a 5-point scale, and smoking and alcohol use were not scored. The subscale score represents the sum of all items. Smoking and alcohol use were nominal categorical variables, reported as adherent or non-adherent.

Self-efficacy was defined as a judgment of one’s own ability to accomplish a certain level of performance [32]. It was measured using the Self-Efficacy for Managing Chronic Diseases (SEMCD) scale, which is commonly used with hypertensive patients [33], and is free and open-access. It comprises six items with an internal consistency reliability of Cronbach’s α = 0.91 [34]; the 10-point scale ranges from “not at all confident” (1) to “totally confident” (10). The total score is the mean of the six items, indicating higher (≥7) or lower (<7) self-efficacy.

Perceived stress was defined as the degree to which life situations are appraised as stressful [35] and measured using the Perceived Stress Scale (PSS), ten items with acceptable reliability at Cronbach’s α = 0.86 [36]. The 5-point Likert response categories range from 0 (never) to 4 (very often), and total scores are tallied by reverse-scoring items 4, 5, 7, and 8, then summing across all ten items. Scores range from low (0–13) to moderate (14–26) to high (27–40). [36]. The scale is free and can be used without permission.

Sociodemographic and clinical data were captured using the case report form. The former include gender, sex, age, ethnicity, race, zip code, education attainment level, marital status, occupation, and insurance/payor. The latter encompass height, weight, BMI, smoking status, number of BP pills, number of BP medications per chart list, number of BP drug classes, hemoglobin A1C, and lipid profile.

Patient experience was defined as the range of interactions with the healthcare system [37] and collected using a 15-item survey. Staff experience was defined as how the employee feels about the job, colleagues, and the organization [38], and was collected using a five-item questionnaire. 

### 2.5. Data Analysis

STATA version 13.1 (StataCorp LLC, College Station, TX, USA) was used as a statistical tool. Descriptive statistics, including frequencies and percentages, were used for nominal- and ordinal-level measurements. PSS was reported using frequencies and percentages. The Wilcoxon signed-rank test was conducted to determine the difference between the pre- and post-intervention rank BP scores. The McNemar test was used to determine the difference in self-efficacy and self-care behaviors pre- and post-intervention. Frequency, percentage, and thematic analyses were used to quantify staff and patient experiences.

## 3. Results

Recruitment spanned ten weeks, enrolling 31 participants with an attrition rate of 13%, for a final sample size of 27. Of the 405 screened, 372 (91.8%) did not meet the inclusion criteria related to age, cellphone or landline access, ability to read, understand, and speak English, or chronic disease. Of the 103 recruited, 72 (69.9%) declined to participate due to the length of the enrollment visit, lack of interest, or the 12-week commitment. Among the 31 who enrolled, one withdrew, and three were lost to follow-up.

### 3.1. Sample Characteristics

Table 1 lists participants’ demographic characteristics. The average participant was middle-aged. Most were female, African-American, single, and employed. They were nearly equally divided between high school graduates and less than high school educated, and either uninsured or covered by Medicaid. Most had an average of two comorbidities that included obesity, dyslipidemia, diabetes, chronic pain, gout, and coronary artery disease. The majority (12, or 44.4%) resided in same zip code as the clinical practice site. 

### 3.2. Blood Pressure Control and Reduction

Blood pressure control at SBP < 130 and DBP < 80 was not achieved, but BP was reduced. The difference in the rank sum distribution of the total scores of SBP and DBP collected at baseline and 12 weeks was statistically significant (Table 2). Mean SBP decreased 13 mm Hg (z = 2.467, *p* = 0.0136), and mean DBP decreased 5 mm Hg (z = 2.593, *p* = 0.0095) from baseline to 12 weeks, with almost similar medians. 

Differences in BP control and reduction aligned with educational attainment and insurance coverage. Those with a high school education or less had worse BP control at baseline, but greater BP reduction at 12 weeks compared to those with a college education. Among insurance coverage groups, uninsured participants’ BP changed little over the intervention, and their control was poorer at 12 weeks than at baseline. Medicare participants had the highest mean SBP reduction at 44 mm Hg, while the combination Medicaid/Medicare group had the highest mean DBP reduction at 21 mm Hg.

### 3.3. Self-Care Behaviors

Statistically significant differences were seen in the H-SCALE subscales of diet (*p* = 0.0001), weight management (*p* = 0.004), and alcohol consumption (*p* = 0.013), while clinically relevant changes were observed in the remaining subscales (see Table 3). Dash diet and weight management adherence increased 22.2% and 7.4.%, respectively, while alcohol abstinence increased 3.7%. Increases were also noted in medication (11.1%), physical activity (29.6%), and smoking adherence (3.7%). 

### 3.4. Self-Efficacy

Self-efficacy did not differ significantly before and after the intervention (*p* = 0.678). However, the percentage of participants with high self-efficacy increased (see Table 4). Specifically, post SECMD scores were higher in 14 participants, lower in 12, and unchanged in one. Among the 14 with higher scores, eight moved from low to high self-efficacy, while six remained unchanged. Among the 14 with higher self-efficacy scores, eight had decreases in both SBP and DBP, while six had a decrease in one or the other. 

### 3.5. Telephone Calls and Home Blood Pressure Measurement

Over the 12-week intervention, 386 calls were attempted, with 101 engaged and completed. Each participant completed an average of 3.74 scheduled calls, with an average duration of 28–29 min. Approximately half of the calls lasted over 40 min, and the other half lasted around 15 min. Although most calls were answered on the first attempt, most participants were not available to engage with and complete the counseling until the third call. 

The 85 scheduled calls that resulted in referral were largely due to participant SBP above 140 mm Hg and DBP above 90 mm Hg, followed by 56 electronic patient portal messages to triage/provider. No scheduled calls with a referral were related to participant-reported stress. 

Participants reported a total of 270 BP readings using HBPM. The majority (176, 65.18%) were categorized as stage II HTN. Only 17 (6.3%) readings were considered normal, 26 (9.63%) were noted as elevated, and 51 (18.89%) fell into the stage I HTN category.

Participants completing more calls had higher BP reductions. Those who completed 1–3 calls had a mean SBP decrease from 155 to 144 mm Hg, and a mean DBP decrease from 92 to 88 mm Hg. Those that completed 4–6 intervention calls had a mean SBP decrease from 151 to 137 mm Hg, and a mean DBP decrease from 94 to 87 mm Hg. 

### 3.6. Clinical Characteristics

As shown in Table 5, BMI remained stable over the 12-week intervention, while lipid profile values increased. The number of self-reported BP medications differed from those prescribed per EMR. The number of participants using ACE inhibitors, angiotensin receptor blockers, aldosterone, and alpha 2 agonists remained the same at baseline and 12 weeks, while those on beta-blockers and calcium channel blockers both increased from nine (33.3%) to 11 (40.7%). Those on diuretics increased from 15 (55.6%) to 19 (70.4%), although loop and thiazide diuretics were not distinguished.

### 3.7. Patient Experience

In the patient satisfaction survey, 26 (96.3%) reported that the intervention was easy to learn and use. All participants understood the feedback about health, and 23 (85.2%) reported that the calls provided enough useful information. Overall, 24 (88.9%) reported that the intervention helped them manage their hypertension. Furthermore, 20 (74.1%) were very satisfied, while six (22.2%) were mostly satisfied with the intervention. No participant responded to the open-ended question “is there anything else you would like to share?”

### 3.8. Staff Experience

The majority of staff (*n* = 9; 81.8%) were satisfied with their experiences with the project. No one indicated dissatisfaction; five shared their opinion about the intervention. In response to the open-ended question asking for additional comments or suggestions, the thematic analysis highlighted teamwork, a favorable view of the intervention, and a favorable view of the counseling material used.

## 4. Discussion

Race, geographic location, educational attainment, and insurance coverage proved relevant social determinants. Most of the sample were African Americans, consistent with site demographics and the literature, which shows racial disparity in HTN prevalence and BP control [39]. Most participants resided in the zip code designated to a poverty priority area, reinforcing the FQHC strategic positioning [40] to address disparities based on location. Consistent with the literature, low educational attainment was associated with high HTN burden and poor BP control [41]. Uninsured participants showed little change in BP over the intervention, which can be explained by their difficulty in accessing healthcare resources. While almost seventy percent of those recruited declined to participate, the sample size was adequate for a feasibility project, as the aim was not to generalize findings. The low-SES population at this FQHC often had other competing priorities that lessened project interest and required time commitment. Stress may also account for this finding: 87.5% of this group reported moderate stress post-intervention, and the literature confirms high levels of stress among the uninsured [42,43].

In this FQHC feasibility project, statistically significant reduction in SBP and DBP (13 and 5 mm Hg, respectively) was consistent with findings from a similar study [44], although the lack of BP control was not. Inability to control BP can be attributed to unrecognized participant stress, medication noncompliance, and provider-prescribing practices. While 78% of the sample reported moderate stress on the PSS, everyone denied having stress during the scripted calls. Prolonged stress can elevate hormone levels, and thus BP and heart rate [35]. There was also a notable discrepancy between the number of self-reported BP meds and the number prescribed per EMR, suggesting participants may not have adhered to prescribed therapies. Findings in low-SES populations indicate that inadequate communication and literacy may be barriers to medication adherence [45]. Additionally, although dual therapy with calcium channel blockers and diuretics is recommended for African Americans [46], it was not predominantly prescribed in our sample. Note that the preference for beta blockers over the dual therapy might be explained by comorbidities, such as coronary heart disease and stroke, which were not among the exclusion criteria.

The difference in adherence to HTN self-care behaviors such as diet, weight management, and alcohol consumption, was statistically significant, consistent with other studies that revealed increased odds of adherence to a low-salt diet and common weight management strategies among African American participants [14]. Although the scripted telephone calls did not address weight management and alcohol consumption, the brochure provided at the baseline office visit describes their effect on BP. These outcomes suggest that provider discussions with this population should target self-care behaviors for HTN management.

Although the difference was not statistically significant, more participants had higher self-efficacy at 12 weeks. As the IMB model and many studies show, higher self-efficacy can be a predictor of self-care behaviors [47,48]. Providing information to participants may promote self-efficacy, increasing their motivation to practice self-care. Interactions between information, motivation, and behavioral skills lead to behavior change and, in this case, BP reduction vs. BP control. 

Higher BP reduction was associated with higher call engagement, possibly due to greater exposure to the intervention. Another study [49] demonstrated increased BP reductions with higher call engagement. 

Poorer lipid profiles and nearly identical BMIs pre- and post-intervention suggest little change in diet, physical activity, or weight management. The project overlapped with the holiday season—Thanksgiving, Christmas, and New Year’s Eve—associated with high cholesterol levels and difficulty with weight loss in previous studies [50,51].

Positive feedback from staff and patient experience surveys, as well as the low attrition rate (12%), demonstrate the intervention’s feasibility. Another study [52] confirms higher patient satisfaction with telehealth interventions than with usual care.

### 4.1. Limitations

The major strengths of this project are that: (a) it is feasible and effective; and (b) it is one of the few telehealth interventions at a FQHC targeting BP control in a low-SES population. Major limitations include its generalizability, the use of convenience sampling, the small sample size, and the use of self-report. As a feasibility and pilot project, it was intended to improve clinical care and bring about immediate positive change in healthcare delivery at the local site. Therefore, results may not be generalizable to all populations. Convenience sampling is known to imply bias and weaken the ability to generalize findings. The small sample size reduces the study’s effect and limits group comparison and evaluation; for example, the predominance of women limited analysis by gender. Given a longer recruitment period, a larger sample would be likely. Last, the HSCALE, PSS, and SEMCD were self-administered, and participants may have answered inappropriately due to a lack of understanding, reducing the internal validity of this pilot study. 

### 4.2. Sustainability

Project sustainability relies on case management, triage, and providers using the script during HTN counseling. Insurance reimbursement would help with Medicaid and Medicare covering brief patient check-ins via communication technologies at an FQHC [53,54]. Favorable project outcomes can be used to support an application to HRSA Telehealth Network Grant Program [55] or the National Institutes of Health funding for the Use of Technology to Enhance Patient Outcomes and Prevent Illness [56]. The FQHC can partner with local higher education institutions to apply for these grants, which would allow it to hire personnel and buy equipment to sustain and expand this program. A partnership with a local university or health department’s health education program could enlist students or health professionals in continuing the intervention.

### 4.3. Recommendations and Implications

Future research should focus on using a randomized controlled trial with an adequate sample size, to confirm the findings of this pilot study in low-SES populations. Other recommendations are to incorporate patient portals that would allow patients to respond at their convenience. Bluetooth technology could be used to collect BP measurements, reducing the self-reporting bias, while promoting engagement and accountability. Last, coronary artery disease and stroke should be additional exclusion criteria to facilitate adequate evaluation of evidence-based prescription practices.

Implications for practice include consideration of using the Office-GAP decision aid tool that incorporates shared decision-making, facilitates medication reconciliation, improves guideline-based prescribing practices, and addresses comorbidities that influence and are influenced by HTN. As for policy, local law-makers should be engaged to advocate for FQHC funding. These institutions are strategically positioned to serve the uninsured and the underserved. They also offer an opportunity for provider and student continuing education in the use of telephone and HBPM interventions as evidence-based practice.

## 5. Conclusions

This innovative project informed the clinical practice site about using telehealth as an effective modality to improve the quality, process, and outcomes of HTN management in a low-SES population. Findings supported this safe, effective, equitable, patient-centered, and timely approach in reducing BP and achieving control, while raising crucial questions about the effects of stress, medication reconciliation and adherence, and guideline-based prescribing. This study contributes to the evidence and practice gap specific to the use of telehealth interventions at this FQHC. 

## Figures and Tables

**Table 1 ijerph-20-05287-t001:** Demographic characteristics of the patient population (*n* = 27).

Characteristics	*n* (%)
Age (Mean age and SD—50 [7.28])	
18–35	2 (7.4)
36–45	4 (14.8)
46–55	21 (77.8)
Gender	
Female	20 (74.1)
Male	7 (25.9)
Race	
White	1 (3.7)
Black/African-American	25 (92.6)
American Indian	1 (3.7)
Education	
High school or less	9 (33.3)
High school graduate	10 (37)
Some college	3 (11.1)
College degree	3 (11.1)
Postbaccalaureate	2 (7.4%)
Marital Status	
Single, never married	13 (48.1)
Married/domestic partnership	3 (11.1)
Separated	1 (3.7)
Divorced	6 (22.2)
Widowed	4 (14.8)
Insurance	
Uninsured	8 (29.6)
Private plan	6 (22.2)
Medicaid	10 (37)
Medicare	2 (7.4)
Combination (Medicaid/Medicare)	1 (3.7)
Occupation	
Employed	16 (59.3)
Unemployed	11 (40.7)

**Table 2 ijerph-20-05287-t002:** Differences in systolic and diastolic blood pressure scores before and 12 weeks after the intervention (*n* = 27).

	*n*	Rank Sum	Average Scores	Medians	Z-Score	*p* ^§^
Systolic Blood Pressure						
Pre-intervention	27	885	153	153	2.467	0.0136
Twelve weeks post-intervention	27	600	140	139		
Diastolic Blood Pressure						
Pre-intervention	27	892	93	94	2.593	0.0095
Twelve weeks post-intervention	27	593	88	90		

Note. ^§^
*p* < 0.05.

**Table 3 ijerph-20-05287-t003:** Hypertension self-care activity level effect at baseline and 12 weeks post-intervention (*n* = 27).

Hypertension Self-Care Activity	Baseline	12 Weeks	
	*n* (%)	*n* (%)	*p* ^§^
Medication			0.839
Medication Adherence	13 (48.2)	16 (59.3)	
Medication Nonadherence	10 (37)	11 (40.7)	
No Medication	4 (14.8)	0 (0)	
Diet			0.0001
Dash Diet Adherence	2 (7.4)	8 (29.6)	
Medium-Quality Diet	10 (37)	8 (29.6)	
Low-Quality Diet	15 (55.6)	11 (40.8)	
Physical Activity			1.000
Physical Activity Adherence	10 (37)	18 (66.67)	
Physical activity Nonadherence	17 (63)	9 (33.33)	
Smoking			0.345
Non-Smoking Adherence	10 (37)	11 (40.7)	
Smoking Non-Adherence	17 (63)	16 (59.3)	
Weight Management			0.004
Weight-Management Adherence	5 (18.5)	7 (25.9)	
Weight-Management NonAdherence	22 (81.5)	20 (74.1)	
Alcohol Consumption			0.013
Abstinence	20 (74.1)	21 (77.8)	
Indulgence	7 (25.9)	6 (22.2)	

Note. ^§^ McNemar test, exact sig. (2-sided); *p* < 0.05.

**Table 4 ijerph-20-05287-t004:** Self-efficacy for managing chronic disease at baseline and 12 weeks post-intervention (*n* = 27).

	Baseline	12 Weeks	
	*n* (%)	*n* (%)	*p* ^§^
High self-efficacy	13 (49.1)	17 (63)	0.678
Low self-efficacy	14 (51.9)	10 (37)	

Note. ^§^ McNemar test, exact sig. (2-sided); *p* < 0.05.

**Table 5 ijerph-20-05287-t005:** Clinical characteristics at baseline and 12 weeks post-intervention: mean and standard deviation (*n* = 27).

	Baseline					12 Weeks				
Characteristics	*n*	M	SD	Min	Max	*n*	M	SD	Min	Max
BMI	27	37.8	15.8	19.9	100.7	27	37.82	15.23	20.02	97
HbA1c	14	8.23	2.84	6	14	16	8.2	2.72	6	14
Blood Glucose	27	144	79.58	59	379	26	148	128.51	74	683
Total Cholesterol	26	169	51.29	22	255	26	180	38.73	86	255
HDL Cholesterol	26	52	14.76	24	86	26	54	15.89	27	99
Triglyceride	26	126	56.62	38	252	26	138	77.14	38	412
LDL Cholesterol	25	102	39.27	36	170	26	113	68.63	29	400
Self-reported BP Med Prescribed	27	1.74	1.09	0	4	27	1.85	0.98	1	4
BP Med per Chart	27	1.81	1.03	0	4	27	2.07	1.03	1	4

Note. BMI = body mass index; HbA1c = hemoglobin A1c; HDL = high-density lipoprotein; LDL = low-density lipoprotein; BP = blood pressure; SD = standard deviation; M = mean; Min = minimum; Max = maximum.

## Data Availability

The data presented in this study can be made available on request.

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
