# Peer review of "Evaluating a Telephone and Home Blood Pressure Monitoring Intervention to Improve Blood Pressure Control and Self-Care Behaviors in Adults with Low-Socioeconomic Status"

_ijerph, 2023, doi:10.3390/ijerph20075287_

Round 1

Reviewer 1 Report

First, for terms that require further explanation, please explain, even briefly. FQHC is one such case. The journal is an international journal, so it is necessary to care about non-US readers. 

Second, why did you exclude adults over the age of 55 from the study? Given that the prevalence of hypertension increases with age, it's hard to argue with the exclusion of older adults, even younger ones. If you could give me the big picture of the entire population of hypertensive patients cared in FQHCs, the setting of this study, I would be able to judge whether your decision was justified. 

Finally, approximately 70% of the recruited study participants declined to participate in the study, resulting in a final study population of 31. If only about 30% of the eligible participated in the study, is the intervention design flawed? I hope you discuss this point in your revised manuscript.

Reviewer 2 Report

Evaluating a Telephone and Home Blood Pressure Monitoring Intervention to Improve Blood Pressure Control and Self-Care Behaviors in Adults with Low-Socioeconomic Status

The study examined the effect of a telehealth intervention on blood pressure control and self-care behaviors in a low socioeconomic status population using a pre-post matched cohort design. Telephone counseling was done every two weeks along with home blood pressure monitoring. Authors reported that, blood pressure control was not achieved, but there was a statistically significant decrease in systolic and diastolic blood pressure. Statistically significant differences were also reported in select self-care behaviors, and greater blood pressure reduction was noted with greater increases in self-efficacy scores and higher call engagement. The intervention was reported to be feasible and effective at reducing blood pressure and increasing self-care behaviors. 

This was a feasibility study and thus understandably has lower sample size. However, authors had given more emphasis on “statistical significance” which is not appropriate. The American Statistical Association (ASA) has released a statement in recent years advising caution when interpreting statistical significance and p-values. The ASA statement emphasized that statistical significance does not imply practical or scientific significance, and that p-values should not be used as a sole decision criterion for scientific conclusions or actions. Authors are advised to tone down emphasis on “statistical significance” throughout the manuscript. 

Also, for showing BP reduction, since non-parametric tests are used, it will be appropriate to show median BP reduction.

While the authors briefly mention some of the limitations of the study, they don't provide a comprehensive discussion of the study's limitations or the generalizability of the findings. Providing a more thorough discussion of these issues would help readers understand the strengths and weaknesses of the study and how the findings might be applicable to other settings or populations.

Round 2

Reviewer 2 Report

Authors have thoroughly revised the manuscript as per suggestion during the previous round of peer review.